# Polyaromatic molecular peanuts

Kohei Yazaki[1,2,†], Munetaka Akita[1], Soumyakanta Prusty[2], Dillip Kumar Chand[2], Takashi Kikuchi[3], Hiroyasu Sato[3] & Michito Yoshizawa[1]

Mimicking biological structures such as fruits and seeds using molecules and molecular assemblies is a great synthetic challenge. Here we report peanut-shaped nanostructures comprising two fullerene molecules fully surrounded by a dumbbell-like polyaromatic shell. The shell derives from a molecular double capsule composed of four W-shaped polyaromatic ligands and three metal ions. Mixing the double capsule with various fullerenes (that is, $C_{60}$, $C_{70}$ and $Sc_3N@C_{80}$) gives rise to the artificial peanuts with lengths of $\sim 3$ nm in quantitative yields through the release of the single metal ion. The rational use of both metal–ligand coordination bonds and aromatic–aromatic $\pi$-stacking interactions as orthogonal chemical glue is essential for the facile preparation of the multicomponent, biomimetic nanoarchitectures.

[1] Laboratory for Chemistry and Life Science, Institute of Innovative Research, Tokyo Institute of Technology, 4259 Nagatsuta, Midori-ku, Yokohama 226-8503, Japan. [2] Department of Chemistry, Indian Institute of Technology Madras, Chennai 600036, India. [3] Rigaku Corporation, 3-9-12 Matsubaracho, Akishima, Tokyo 196-8666, Japan. † Present address: Faculty of Engineering, Yamanashi University, 4-3-11, Takeda, Kofu-shi 400-8511, Japan. Correspondence and requests for materials should be addressed to M.Y. (email: yoshizawa.m.ac@m.titech.ac.jp).

Mimicking the fascinating shapes and functions of biological structures using simple molecules and artificial molecular assemblies is an ongoing challenge for synthetic chemists[1]. Mechanical bio-motions such as shuttling and rotation have been successfully imitated by interlocked supramolecules and stimuli-responsive molecules[2–8]. On the other hand, multilayered, multicomponent bio-structures such as fruits and seeds are so complicated that chemical mimicry of such structures is extremely difficult[9–12]. Peanuts are well known and relatively simple seeds comprising a couple of beans fully enclosed by a pod (Fig. 1a)[13]. There have been several synthetic reports on multicomponent nanostructures with two or three open cavities capable of binding relatively small ions and metal complexes, such as $Cl^-$, $PF_6^-$ and cisplatin[14–18]. However, imitation of the characteristic core–shell structures has yet to be accomplished on the nanoscale.

Here we report the rational design and facile synthesis of peanut-shaped nanostructures constructed of polyaromatic frameworks. Our synthetic approach to the molecular peanuts is the use of both metal–ligand coordination bonds[19–22] and aromatic–aromatic π-stacking interactions[23,24] as orthogonal chemical 'glue' to connect multiple molecular components. For the precursor to the dumbbell-shaped polyaromatic 'pod', we first design a W-shaped polyaromatic ligand, which binds with metal ions through coordination bonds to form a molecular double capsule with two spherical cavities (Fig. 1b, step i). Next, efficient π-stacking interactions between the two polyaromatic cavities and two spherical fullerene 'beans' drive the formation of a molecular peanut (Fig. 1b, step ii). This step is accompanied by the release of the central metal hinge from the double capsule due to steric repulsion. We also report that the two closed cavities of the double capsule encapsulate two different, medium-sized molecules in a heterolytic manner.

## Results

**Design of a molecular peanut**. To construct a peanut-shaped nanostructure, we specifically designed W-shaped tripyridine ligand **1** containing four anthracene rings and two *meta*-phenylene spacers bearing two or three hydrophilic methoxyethoxy pendants (Fig. 1c), on the basis of our previous synthesis of $M_2L_4$ polyaromatic capsules[25–30]. We expected that the W-shaped polyaromatic ligands assemble into an $M_3L_4$ double capsule (**2**) upon complexation with square-planar metal ions. In addition, the double capsule converts to molecular peanut assemblies (**G$_2$@3**) with lengths of approximately 3 nm upon encapsulation of spherical polyaromatic compounds (that is, **G** = fullerenes $C_{60}$ and $C_{70}$ and metallofullerene $Sc_3N@C_{80}$) through demetallation from the central pyridine rings. The first and second steps occur spontaneously and quantitatively under the control of designed coordinative and π-stacking interactions, respectively. It should be noted that W-shaped ligand **1** exists as a mixture of 10 stereoisomers (see Supplementary Fig. 1) due to restricted rotation about the sterically hindered, pyridyl–anthryl and anthryl–phenyl bonds[25,31]. Thus double capsule **2** is favoured thermodynamically over possible $>10^3$ $M_3L_4$ isomers.

**Quantitative formation of double capsules**. W-shaped polyaromatic ligand **1a** was prepared from a *meta*-bis(10-bromo-

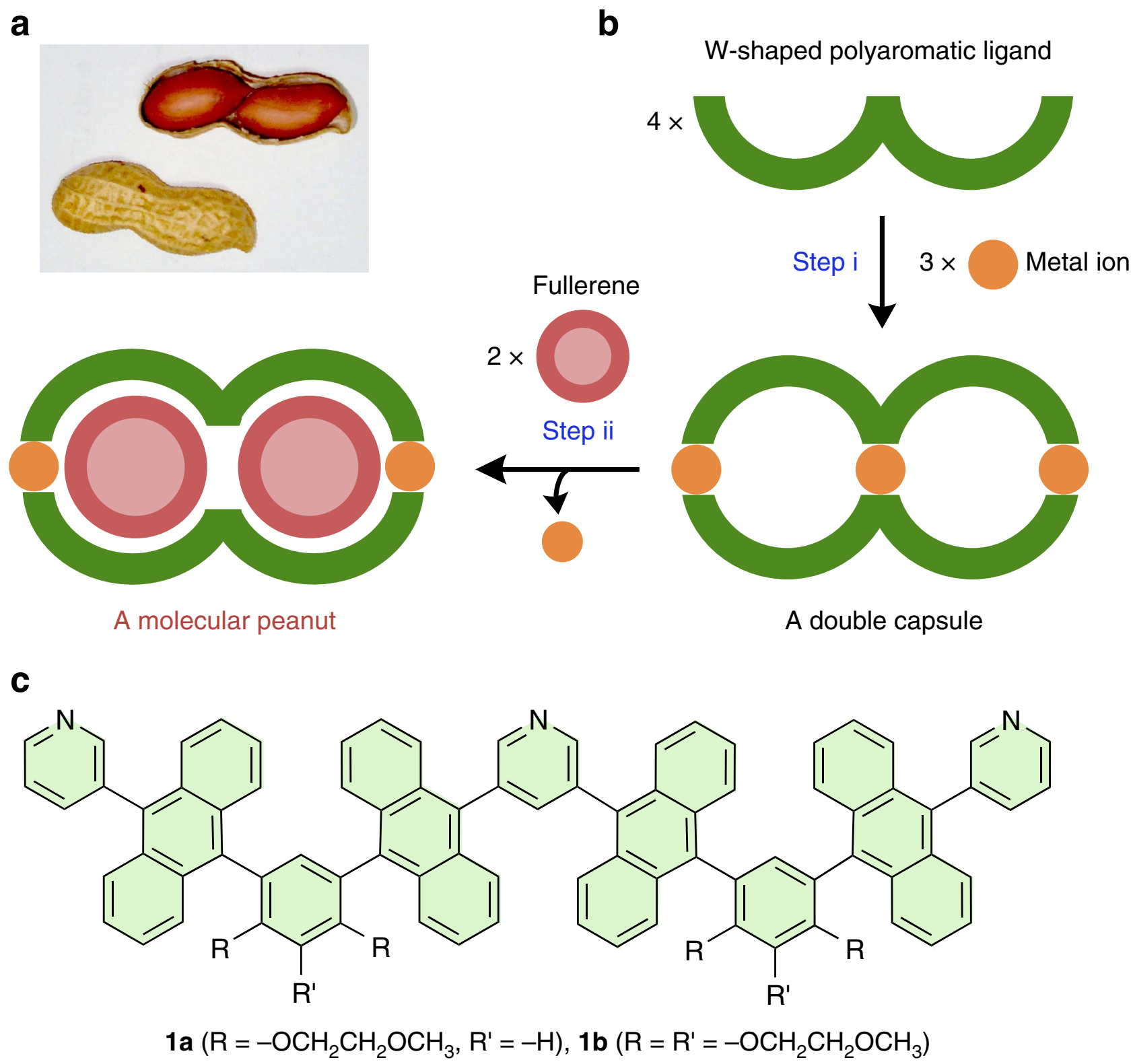


**Figure 1 | Design and synthetic strategy of a molecular peanut.** (**a**) Photograph of a peanut. (**b**) Schematic representation of the stepwise formation of (i) a molecular double capsule from W-shaped polyaromatic ligands and metal ions using coordination bonds and (ii) a molecular peanut from the double capsule and fullerene molecules using π-stacking interactions. (**c**) W-shaped polyaromatic ligands **1a** and **1b** designed herein.

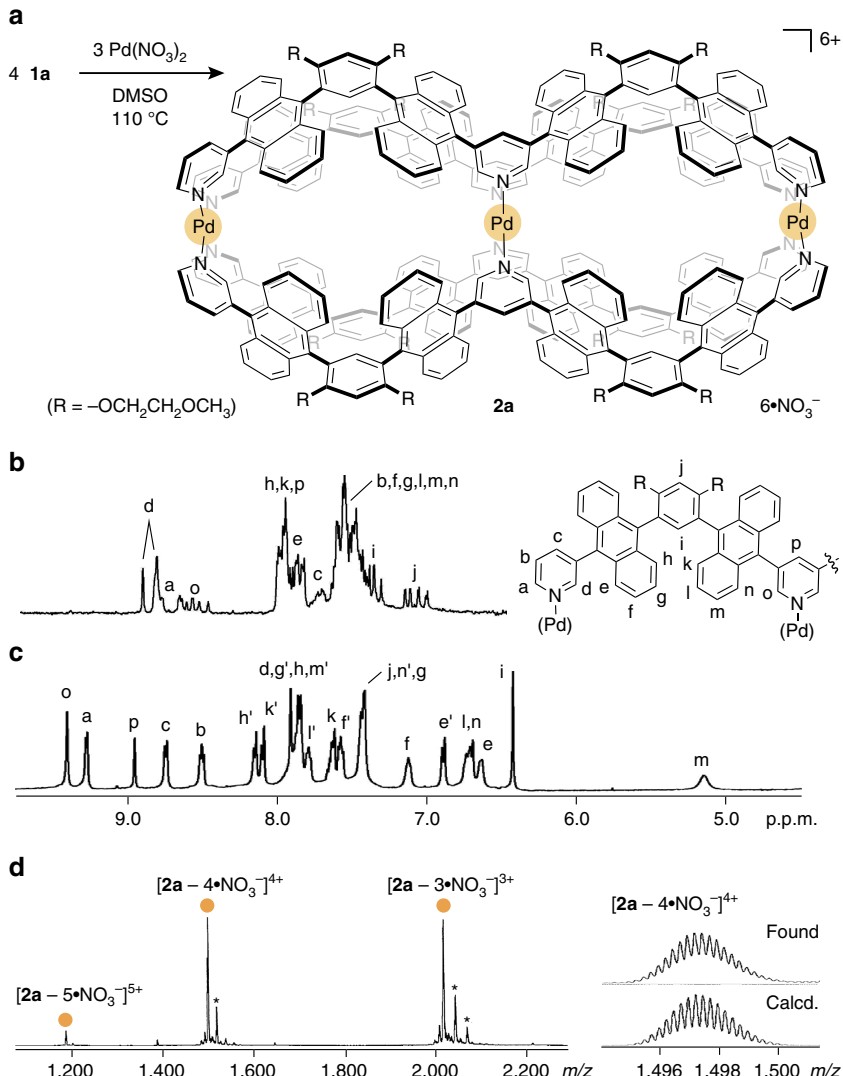

**Figure 2 | Quantitative formation and characterization of a double capsule.** (**a**) Schematic representation of the formation of double capsule **2a**. (**b**) $^1$H NMR spectra (500 Hz, DMSO-$d_6$, room temperature) of an isomeric mixture of ligand **1a** and (**c**) double capsule **2a**. (**d**) ESI-TOF MS spectrum (CH$_3$OH) of **2a** and the expansion and simulation of the [**2a** − 4•NO$_3^-$]$^{4+}$ signal (the DMSO adducts are marked with asterisks).

9-anthryl)benzene derivative[25] by two-step Suzuki-Miyaura cross-coupling reactions in 51% yield (see Supplementary Methods). The matrix-assisted laser desorption ionization time-of-flight mass spectrometry (MALDI-TOF MS) spectrum showed a single peak at $m/z = 1,386.2$, corresponding to **1a**. In contrast, a complicated array of signals was observed in the $^1$H NMR spectrum (Fig. 2b) due to the presence of numerous stereoisomers of **1a** (see Supplementary Fig. 1). The energies of 10 stereoisomers of **1a′** (R = -OCH$_3$) are comparable ($\Delta E < 0.1$ kcal mol$^{-1}$) in the ground state, as indicated by semiempirical calculations. When ligand **1a** (1.7 µmol) was combined with Pd(NO$_3$)$_2$ (1.8 µmol) in dimethylsulfoxide (DMSO)-$d_6$ (0.5 ml) at 110 °C for 12 h, double capsule **2a** was formed quantitatively (Fig. 2a). The $^1$H NMR spectrum of the product exhibited relatively simple signals (Fig. 2c), confirming the conversion of ligand **1a** into a highly symmetrical assembly. One set of pyridyl signals $H_{a,b,c}$ and $H_{o,p}$ is shifted downfield between 9.36 and 8.47 p.p.m., implying the formation of coordinative pyridyl-Pd(II) bonds. The proton signals of the phenylene moieties ($H_i$) and the anthracene moieties ($H_m$) are shifted upfield ($\Delta\delta = -0.93$ and $-2.35$ p.p.m., respectively) due to efficient aromatic shielding, which indicates the formation of

inner cavities defined by tightly packed polyaromatic frameworks. The electron-spray ionization (ESI)-TOF MS analysis confirmed the formation of M$_3$L$_4$ assembly **2a** with a molecular weight of 6231.86 Da. Prominent molecular ion peaks were observed at $m/z = 1,497.2$ and 2,016.9 assignable to the [**2a**–4•NO$_3^-$]$^{4+}$ and [**2a**–3•NO$_3^-$]$^{3+}$ species, respectively (Fig. 2d).

Unambiguous structural evidence of the M$_3$L$_4$ double capsule was provided by single crystal X-ray diffraction analysis. Pale yellow crystals suitable for X-ray analysis grew upon slow diffusion of tetrahydrofuran and diethyl ether into a DMSO solution of **2a′** (the BF$_4^-$ analogue of **2a**) at room temperature for 1 week. The molecular structure of the double capsule reveals a 3.2 nm long dumbbell-shaped framework consisting of two polyaromatic spheres linked together (Fig. 3a,b). The shape closely resembles fullerene dimer C$_{120}$ with a length of 1.6 nm (ref. 32). The distance between the two terminal Pd(II) hinges is 2.8 nm and each of the cavity diameter is ~1.1 nm. Average dihedral angles between the pyridine rings and the nearby anthracene rings (77.0° and 62.0° at the central and terminal parts, respectively) indicate structural strain around the crowded, central pyridine rings. The structure has two identical cavities with an average volume of 500 Å$^3$, each fully encircled by

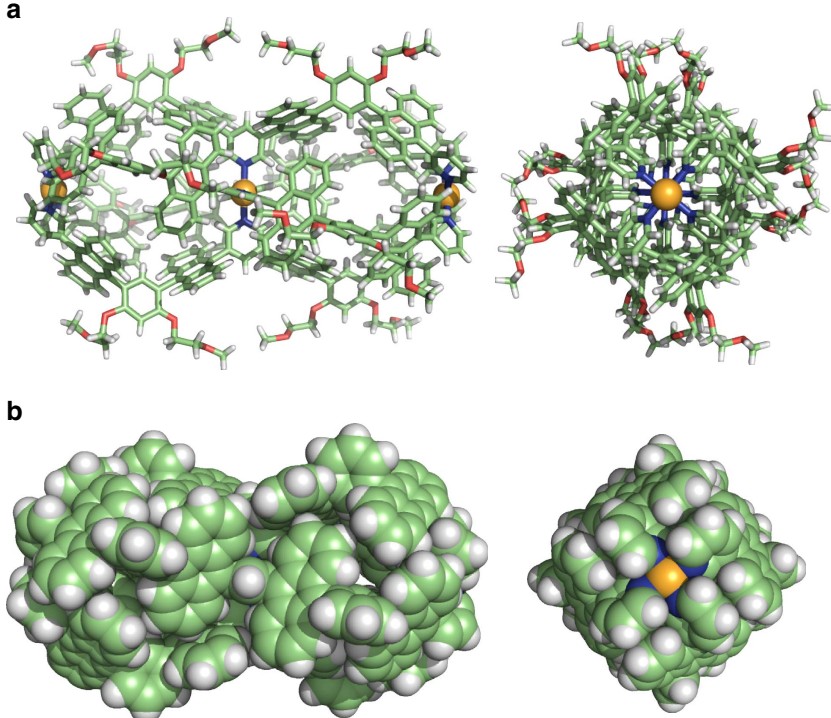

**Figure 3 | Crystal structures of double capsule 2a′.** (**a**) The ball-and-stick representation of double capsule **2a′**, which is the $BF_4^-$ analogue of **2a**, (counterions and solvents are omitted for clarity) and (**b**) its space-filling representation (the peripheral substituents are replaced by hydrogen atoms).

the eight anthracene panels in a twisted conformation (Fig. 3b). The flexible, 16 methoxyethoxy substituents assist in dissolution of the rigid polyaromatic shell of **2a** in polar organic solvents. The solubility could be improved by further attachment of methoxyethoxy groups on the *meta*-phenylene spacers. Double capsule analogue **2b** with 24 methoxyethoxy groups, prepared from ligand **1b** (Fig. 1c) and Pd(II) ions (see Supplementary Methods), was soluble in DMSO ( > 60 mM), $CH_3CN$ ( > 30 mM) and 100:1 $H_2O/CH_3CN$ ( ∼ 30 μM) solutions (see Supplementary Fig. 27).

**Quantitative formation of molecular peanuts**. The conversion from double capsule **2b** to molecular peanuts occurred quantitatively upon treatment with various fullerenes in hot DMSO solution. For example, when black $C_{60}$ powder (7.0 equiv. based on **2b**) suspended in a DMSO-$d_6$ solution (0.4 ml) of **2b** (0.1 μmol) was heated at 110 °C for 1 night, the colour of the solution changed from pale yellow to red, indicating the formation of a $(C_{60})_2$●**3b** structure (Fig. 4a). In the $^1H$ NMR spectrum, the aromatic signals of **2b** fully converted to a single set of new signals (Fig. 4b). Notably, pyridine signals $H_o$ and $H_p$ were observed at 8.68 and 7.80 p.p.m., respectively, with large upfield shifts ($\Delta\delta_{max} = -1.01$ p.p.m.) as compared with those of **2b**. The shifts suggest the removal of the central Pd(II) ion through the cleavage of the Pd(II)-pyridine coordination bonds. In contrast, the two terminal Pd(II) ions remained bound as indicated by the lack of changes for pyrdine signals $H_{a-c}$. The $^{13}C$ NMR spectrum showed a single prominent signal for $C_{60}$ at 139.3 p.p.m. (see Supplementary Fig. 36). The upfield shift ($\Delta\delta = -3.4$ p.p.m.) relative to the carbon signal of free $C_{60}$ in $CDCl_3$ and the intensity support encapsulated fullerenes. The $^1H$ diffusion-ordered spectroscopy (DOSY) NMR spectrum showed a single band with a diffusion coefficient (*D*) of $6.31 \times 10^{-11}$ $m^2$ $s^{-1}$ (see Supplementary Fig. 39), which

indicates the product size being approximately 3.4 nm (sphere model). The ESI-TOF MS analysis definitely confirmed the $M_2L_4$●$(C_{60})_2$ composition with a molecular weight of 8,041.44 Da. The main peaks at $m/z = 1,948.1$ and 2,618.1 are assignable to $[(C_{60})_2@\mathbf{3b} - n\bullet NO_3^-]^{n+}$ species ($n = 4$ and 3, respectively) (Fig. 4d).

The optimized structure of $(C_{60})_2@\mathbf{3b}$ (R = − H) by force-field calculations, on the basis of the detailed NMR and MS analyses, shows a peanut-shaped nanostructure (Fig. 4e), where two spherical $C_{60}$ 'beans' (1.0 nm van der Waals diameter) accommodated in the cavity are fully covered with the dumbbell-like polyaromatic 'pod' of **3b**. The closest distance between the two fullerene guests is 7.1 Å. The fullerenes effectively fill the capsule cavity and form extensive aromatic contacts with 16 anthracene panels of **3b** ( < 3.4 Å). The relatively high stability of $(C_{60})_2@\mathbf{3b}$ under high dilution conditions ( ∼ 5 μM in DMSO) (see Supplementary Fig. 34e,f) suggests that multiple π-stacking interactions between the host and guests stabilize the complex and most likely play a major role in binding (see Supplementary Fig. 63). Heating is essential for the $(C_{60})_2@\mathbf{3b}$ formation. Presumably the elevated temperature helps the dissociation of the central Pd(II) ion, partial or full capsule disassembly and the dissolution of $C_{60}$ in DMSO. It is noteworthy that molecular peanut $(C_{60})_2@\mathbf{3b}$ could also be obtained quantitatively in one step by mixing ligand **1b** and $Pd(NO_3)_2$ (in a 2:1 ratio) and excess $C_{60}$ in DMSO at 110 °C for 12 h (Fig. 4a and see Supplementary Fig. 34c,d).

Polyaromatic molecular peanuts possessing higher fullerenes $C_{70}$ and metallofullerenes $Sc_3N@C_{80}$ were also obtained exclusively by the same one-pot reaction in DMSO. The $^1H$ NMR and ESI-TOF MS analyses provided the structural evidences of 1:2 host–guest complexes $(C_{70})_2@\mathbf{3b}$ and $(Sc_3N@C_{80})_2@\mathbf{3b}$. For example, the proton NMR pattern of $(Sc_3N@C_{80})_2@\mathbf{3b}$ is similar to that of $(C_{60})_2@\mathbf{3b}$ except for inner phenylene signals $H_i$ at 6.84 p.p.m. (Fig. 4c). The product composition of a $\mathbf{3b}●(Sc_3N@C_{80})_2$ with a large molecular weight (8,819.03 Da) was unequivocally identified by the ESI-TOF

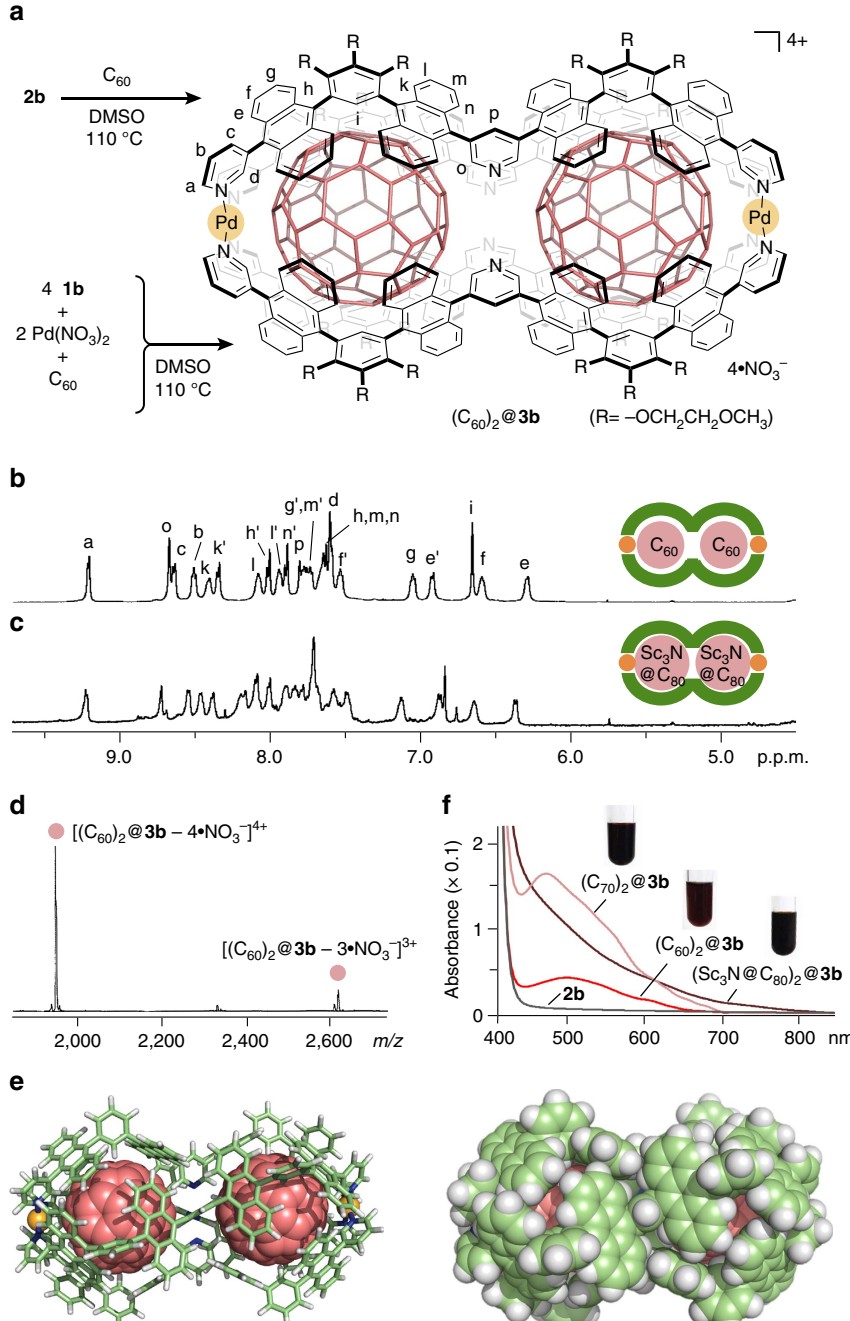

**Figure 4 | Quantitative formation and characterization of molecular peanuts. (a)** Schematic representation of the formation of polyaromatic molecular peanut $(C_{60})_2$@**3b**. $^1$H-NMR spectra (500 MHz, DMSO-$d_6$, room temperature) of (**b**) $(C_{60})_2$@**3b** and (**c**) $(Sc_3N@C_{80})_2$@**3b**. (**d**) ESI-TOF MS spectrum (DMSO) of $(C_{60})_2$@**3b**. (**e**) Optimized structures of molecular peanut $(C_{60})_2$@**3b** (R = − H; ball-and-stick and space-filling models, the peripheral substituents are replaced by hydrogen atoms for clarity). (**f**) Ultraviolet–visible spectra and photographs (DMSO, room temperature) of $(C_{60})_2$@**3b**, $(C_{70})_2$@**3b**, $(Sc_3N@C_{80})_2$@**3b** and **2b**.

MS analysis (see Supplementary Fig. 48). The characteristic absorption bands derived from the accommodated fullerenes in **3b** were clearly observed in the ultraviolet–visible spectra (Fig. 4f). The DMSO solutions of $(C_{60})_2$@**3b** and $(C_{70})_2$@**3b** showed broad absorption bands at $\lambda_{max} = 501$ and 476 nm, respectively, and that of $(Sc_3N@C_{80})_2$@**3b** exhibited a shoulder band around 500 nm. In the optimized structure of $(Sc_3N@C_{80})_2$@**3b** (R = − H), two molecules of the large spherical metallofullerene (1.1 nm van der Waals diameter) are again fully wrapped by the polyaromatic shell (see Supplementary Fig. 61). This intriguing core–shell–shell nanostructure remains intact even after several days under ambient conditions.

**Selective formation of a heteroleptic complex.** Selective heteroleptic encapsulation of medium-sized, aliphatic and aromatic compounds was observed within double capsule **2b**. When excess amounts of hydrophobic diamantane (**4a**) and phenanthrene (**4b**) (100 equiv. each based on **2b**) were suspended in a 100:1 $D_2O/CD_3CN$ solution of **2b** at 60 °C for 3 h, the two hydrophobic cavities of **2b** separately bound one molecule of **4a** and two molecules of **4b** to generate 1:1:2 host–guest–guest' complex (**4a**/(**4b**)$_2$)@**2b** in nearly quantitative yield (Fig. 5a,b). The $^1$H NMR spectrum of the product clearly showed new signals derived from the encapsulated **4a** in the range of − 1.75 to − 1.32 p.p.m.

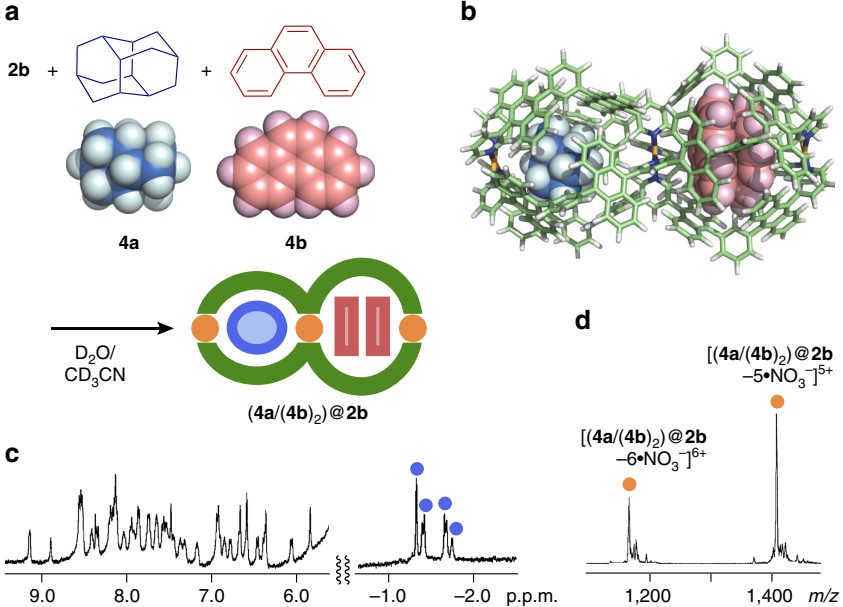

**Figure 5 | Selective formation and characterization of a heteroleptic complex.** (**a**) Schematic representation of the selective formation of heteroleptic host-guest-guest' complex (**4a**/(**4b**)$_2$)@**2b** and (**b**) its optimized structure (the peripheral substituents are replaced by hydrogen atoms). (**c**) $^1$H NMR (500 MHz, 100:1 D$_2$O/CD$_3$CN, room temperature) and (**d**) ESI-TOF MS (H$_2$O/CH$_3$CN) spectra of (**4a**/(**4b**)$_2$)@**2b**.

(Fig. 5c). The selective formation of a **2b**•**4a**•(**4b**)$_2$ composite was confirmed by the ESI-TOF MS spectrum displaying prominent peaks at $m/z = 1,412.9$ and 1,781.6 (Fig. 5d). The NMR and MS spectra of the heteroleptic (**4a**/(**4b**)$_2$)@**2b** complex are quite different from the spectra of homoleptic host–guest complexes (**4a**/**4a**)@**2b** and (**4b**/(**4b**)$_2$)@**2b**, independently prepared from **2b** with **4a** or **4b**, respectively, under similar aqueous conditions (see Supplementary Methods). ESI-TOF MS spectra of (**4a**/**4a**)@**2b** and (**4b**/(**4b**)$_2$)@**2b** showed prominent peaks at, for example, $m/z = 1,379.3$ [(**4a**/**4a**)@**2b** – 5•NO$_3^-$]$^{5+}$ and 1,410.9 [(**4b**/(**4b**)$_2$)@**2b** – 5•NO$_3^-$]$^{5+}$, respectively (see Supplementary Figs 52 and 57).

The unusual heteroleptic encapsulation most probably arises from the cooperative changes in volume of the linked cavities upon guest encapsulation, which is mechanistically different from a previous heteroleptic binding in three open cavities with different binding sites[18]. Optimized host–guest structures of **2b′** (R = −H) using force-field calculations showed that the volume of the second cavity increases by 6% to 530 Å$^3$ upon encapsulation of aliphatic guest **4a** (210 Å$^3$) in the first cavity (see Supplementary Fig. 62). Correspondingly, the volume of the second cavity decreases slightly (−4%) upon encapsulation of aromatic guests (**4b**)$_2$ (total 400 Å$^3$) in the first cavity. This heteroleptic encapsulation is also in sharp contrast to the pairwise encapsulation of two different guests in a single host cavity[28,33–37].

## Discussion
We have created peanut-shaped polyaromatic nanostructures through stepwise and one-pot multicomponent self-assembly. The structures are composed of a dumbbell-like polyaromatic shell with metal hinges and two spherical polyaromatic molecules, such as fullerene C$_{60}$ and metallofullerene Sc$_3$N@C$_{80}$. The unusual core–shell nanostructures, with lengths of approximately 3 nm and molecular weights of up to ∼8,820 Da, quantitatively form through multiple coordination bonds and π-stacking interactions as orthogonal chemical glue. This simple synthetic strategy should provide wide-ranging utilities for the facile preparation of advanced artificial nanoarchitectures inspired by complex natural systems.

## Methods

**General.** NMR: Bruker AVANCE-HD500 (500 MHz), ESI-TOF MS: Bruker micrOTOF II, ultraviolet–visible: JASCO V-670DS, FT-IR: JASCO FT/IR-4200, X-ray: Rigaku XtaLAB Pro P200, Elemental analysis: LECO CHNS-932 VTF-900, Molecular Force-field Calculation: Materials Studio (version 5.5.3, Accelrys Software Inc.). Solvents and reagents: TCI Co., Ltd., Wako Pure Chemical Industries Ltd., Kanto Chemical Co., Inc., Sigma-Aldrich Co., and Cambridge Isotope Laboratories, Inc. Compounds **6a,b** and their precursors (**5a,b**) (see Supplementary Methods and Supplementary Figs 2–6 and 19–22) were synthesized according to previously reported procedures[25,38].

**Synthesis of ligand 1a.** Compound **6a** (0.968 g, 1.32 mmol), 3,5-pyridinediboronic acid bis(pinacol) ester (0.229 g, 0.690 mmol), K$_3$PO$_4$ (1.50 g, 7.07 mmol) and Pd(PPh$_3$)$_4$ (113 mg, 97.4 μmol) were added to a 50 ml glass flask filled with N$_2$. Dry dimethylformamide (30 ml) was added to the flask and then the mixture was stirred at 80 °C for 2 days. The resultant solution was concentrated under reduced pressure. After addition of water, the crude product was extracted with CHCl$_3$. The obtained organic layer was dried over MgSO$_4$, filtrated and concentrated under reduced pressure. The crude product was purified by gel permeation chromatography to afford **1a** as a yellow powder (0.467 g, 0.337 mmol, 51% yield) (see Supplementary Figs 7–10b). Ligand **1b** was also prepared by the same way (see Supplementary Methods and Supplementary Figs 23–26).

$^1$H NMR (500 MHz, CDCl$_3$, room temperature): $\delta$ 8.94–8.58 (m, 6H), 8.06–7.29 (m, 39H), 7.18–7.11 (m, 2H), 4.17–4.12 (m, 8H), 3.43–3.32 (m, 8H), 2.99–2.77 (m, 12H). $^{13}$C NMR (125 MHz, CDCl$_3$, room temperature): $\delta$ 158.5 (C$_q$), 152.0 (CH), 151.2 (CH), 149.0 (CH), 142.1 (CH), 139.6 (CH), 137.1 (CH), 135.2 (C$_q$), 134.8 (C$_q$), 134.6 (C$_q$), 132.6 (C$_q$), 132.3 (C$_q$), 130.5 (C$_q$), 130.4 (C$_q$), 127.5 (CH), 127.4 (CH), 126.4 (CH), 125.8 (CH), 125.6 (CH), 125.2 (CH), 123.5 (CH), 120.5 (CH), 100.2 (CH), 71.0 (CH$_2$), 69.3 (CH$_2$), 59.1 (CH$_3$). FT-IR (KBr, cm$^{-1}$): 3,022; 2,926; 2,877; 2,819; 1,505; 1,484; 1,377; 1,313; 1,265; 1,191; 1,170; 1,152; 1,128; 1,103; 1,025; 766, 688, 609. MALDI-TOF MS (dithranol): $m/z$ Calcd. for C$_{95}$H$_{75}$N$_3$O$_8$: 1,386.56, Found 1,386.24 [M]$^+$. HR MS (ESI, CH$_2$Cl$_2$/CH$_3$OH): $m/z$ Calcd. for C$_{95}$H$_{75}$N$_3$O$_8$: 1,387.5660, Found 1,387.5662 [M + H]$^+$.

**Formation of double capsule 2a.** W-shaped ligand **1a** (2.4 mg, 1.7 μmol), a DMSO-$d_6$ solution (25 mM) of Pd(NO$_3$)$_2$ (70 μl, 1.8 μmol), which prepared *in situ* from PdCl$_2$(DMSO)$_2$ and AgNO$_3$, and DMSO-$d_6$ (0.5 ml) were added to a glass test tube and then the mixture was stirred at 110 °C for 12 h. The quantitative formation of double capsule **2a** was confirmed by NMR, MS and X-ray crystallographic analyses (see Supplementary Figs 11–18). The diffusion coefficient (*D*) of **2a** in DMSO-$d_6$ was estimated to be $6.31 \times 10^{-11}$ by the $^1$H DOSY NMR analysis, which indicates the formation of a 3.4 nm-sized structure (sphere model). Double capsule **2a′**, which is a BF$_4^-$ analogue of **2a**, was also obtained by using Pd(BF$_4$)$_2$ (see Supplementary Fig. 11b). The single crystals for X-ray crystallographic analysis were obtained by slow diffusion of tetrahydrofuran and diethyl ether into a DMSO solution of **2a′** at room temperature for a week (see Supplementary Methods, Supplementary Table 1 and

Supplementary Fig. 60). Double capsule **2b** was prepared by the same way (see Supplementary Methods and Supplementary Figs 27–33).

$^{1}$H NMR (500 MHz, DMSO-$d_6$, room temperature): $\delta$ 9.36 (s, 2H), 9.24 (d, $J = 5.5$ Hz, 2H), 8.92 (s, 1H), 8.71 (d, $J = 7.5$ Hz, 2H), 8.47 (dd, $J = 7.5$, 5.5 Hz, 2H), 8.12 (d, $J = 8.5$ Hz, 2H), 8.06 (d, $J = 8.5$ Hz, 2H), 7.88 (s, 2H), 7.83–7.76 (m, 8H), 7.63–7.53 (m, 4H), 7.41–7.39 (m, 6H), 7.10 (br, 2H), 6.87 (d, $J = 8.5$ Hz, 2H), 6.72-6.67 (m, 4H), 6.61 (br, 2H), 6.41 (s, 2H), 5.19 (br, 2H), 4.37–4.18 (m, 8H), 4.36–3.14 (m, 8H), 2.99–2.76 (m, 12H). $^{13}$C NMR (125 MHz, DMSO-$d_6$, room temperature): $\delta$ 158.3 ($C_q$), 158.0 ($C_q$), 152.7 (CH), 152.3 (CH), 151.0 (CH), 147.4 (CH), 144.2 (CH), 139.0 ($C_q$), 136.6 ($C_q$), 136.1 ($C_q$), 135.5 (CH), 135.3 ($C_q$), 132.1 ($C_q$), 131.6 (CH), 131.5 ($C_q$), 129.6–123.9, 117.9 ($C_q$), 117.7 ($C_q$), 99.8 (CH), 70.2 (CH$_2$), 70.0 (CH$_2$), 68.2 (CH$_2$), 67.9 (CH$_2$), 58.0 (CH$_3$). DOSY NMR (500 MHz, DMSO-$d_6$, 298 K): $D = 6.31 \times 10^{-11}$ m$^2$ s$^{-1}$. FT-IR (KBr, cm$^{-1}$): 3,061; 2,926; 2,880; 2,817; 1,602; 1,504; 1,440; 1,379; 1,315; 1,267; 1,192; 1,126; 1,104; 1,028; 944, 769, 687. ESI-TOF MS (CH$_3$CN): $m/z$ 1,185.4 [**2a** $-$ 5•NO$_3^-$]$^{5+}$, 1,497.2 [**2a** $-$ 4•NO$_3^-$]$^{4+}$, 2,016.9 [**2a** $-$ 3•NO$_3^-$]$^{3+}$.

**X-ray crystal data of 2a′.** C$_{380}$H$_{300}$BF$_4$N$_{12}$O$_{32}$Pd$_3$, $M_r = 5{,}952.61$, tetragonal, $P4/ncc$, $a = b = 26.5390(8)$ Å, $c = 71.806(7)$ Å, $V = 50{,}574(5)$ Å$^3$, $Z = 4$, $\rho_{calcd} = 0.782$ g cm$^{-3}$, $F(000) = 12{,}396$, $T = 93$ K, reflections collected/unique 152,900/25,558 ($R_{int} = 0.0690$), $R_1 = 0.1060$ ($I > 2\sigma(I)$), $wR_2 = 0.3855$, GOF = 1.204. All the diffraction data were collected on a diffractometer ($\lambda$(CuK$\alpha$) = 1.54187 Å). The contribution of the electron density associated with greatly disordered counterions and solvent molecules, which could not be modelled with discrete atomic positions, were handled using the SQUEEZE routine in PLATON.

**Formation of molecular peanut (C$_{60}$)$_2$@3b.** Route 1: Double capsule **2b** (0.7 mg, 0.1 μmol), fullerene C$_{60}$ (0.5 mg, 0.7 μmol) and DMSO-$d_6$ (0.4 ml) were added to a glass test tube and then the mixture was stirred at 110 °C for 12 h. The quantitative formation of (C$_{60}$)$_2$@**3b** was confirmed by NMR, MS and ultraviolet–visible analyses. Route 2: A DMSO-$d_6$ solution (50 mM) of Pd(NO$_3$)$_2$ (42 μl, 2.1 μmol), ligand **1b** (6.0 mg, 3.9 μmol), fullerene C$_{60}$ (3.0 mg, 4.2 μmol) and DMSO-$d_6$ (0.5 ml) were added to a glass test tube and then the mixture was stirred at 110 °C for 6 h. The quantitative formation of (C$_{60}$)$_2$@**3b** was confirmed by NMR, MS and ultraviolet–visible analyses (see Supplementary Figs 34,36–40a and 49). Host–guest complexes (C$_{60}$)$_2$@**3a**, (C$_{70}$)$_2$@**3b** and (Sc$_3$N@C$_{80}$)$_2$@**3b** were obtained quantitatively by the same way (see Supplementary Figs 35 and 40b–49).

$^{1}$H NMR (500 MHz, DMSO-$d_6$, room temperature): $\delta$ 9.21 (d, $J = 6.0$ Hz, 2H), 8.68 (s, 2H), 8.64 (d, $J = 6.5$ Hz, 2H), 8.51 (dd, $J = 6.5$, 6.0 Hz, 2H), 8.41 (br, 2H), 8.35 (d, $J = 8.0$ Hz, 2H), 8.09 (br, 2H), 8.02 (d, $J = 8.5$ Hz, 2H), 7.95 (br, 2H), 7.90 (d, $J = 9.0$ Hz, 2H), 7.81–7.73 (m, 5H), 7.65–7.59 (m, 8H), 7.54 (br, 2H), 7.06 (br, 2H), 6.93 (d, $J = 7.5$ Hz, 2H), 6.66 (s, 2H), 6.60 (br, 2H), 6.30 (d, $J = 7.0$ Hz, 2H), 4.41–4.31 (m, 4H), 4.00–3.90 (m, 4H), 3.85–3.79 (m, 4H), 3.74 (t, $J = 4.5$ Hz, 4H), 3.33 (s, 6H), 3.08 (t, $J = 5.0$ Hz, 4H), 2.87 (t, $J = 4.0$ Hz, 4H), 2.69 (s, 6H), 2.55 (s, 6H). ESI-TOF MS (CH$_3$CN) of (C$_{60}$)$_2$@**3b**: $m/z$ 1,948.1 [(C$_{60}$)$_2$@**3b** $-$ 4•NO$_3^-$]$^{4+}$, 2,618.1 [(C$_{60}$)$_2$@**3b** $-$ 3•NO$_3^-$]$^{3+}$.

**Formation of heteroleptic complex (4a/(4b)$_2$)@2b.** Diamantane (**4a**; 0.7 mg, 3.7 μmol) and phenanthrene (**4b**; 0.5 mg, 2.8 μmol) were added to 100:1 D$_2$O:CD$_3$CN solution (0.6 ml) of double capsule **2b** (1.7 mg, 0.22 μmol) and the mixture was stirred at room temperature for 3 h. The quantitative formation of a (**4a**/(**4b**)$_2$)@**2b** complex was confirmed by NMR and ESI-TOF MS analyses (see Supplementary Figs 58,59 and 62).

ESI-TOF MS (H$_2$O:CH$_3$CN = 100:1): $m/z$ 1,166.9 [(**4a**/(**4b**)$_2$)@**2b** $-$6•NO$_3^-$]$^{6+}$, 1,412.9 [(**4a**/(**4b**)$_2$)@**2b** $-$ 5•NO$_3^-$]$^{5+}$, 1,781.6 [(**4a**/(**4b**)$_2$)@**2b**$-$4•NO$_3^-$]$^{4+}$, 2,396.1 [(**4a**/(**4b**)$_2$)@**2b** $-$ 3•NO$_3^-$]$^{3+}$.

**Data availability.** The authors declare that the data supporting the findings of this study are available within the Supplementary Information files and from the corresponding author upon reasonable request. CCDC-529139 contains the supplementary crystallographic data for the structure reported in this article. The data can be obtained free of charge from The Cambridge Crystallographic Data Centre (CCDC) via www.ccdc.cam.ac.uk/data_request/cif.

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

## Acknowledgements

This work was supported by JSPS KAKENHI (Grant No. JP25104011/JP26288033/JP17H05359), SERB, Department of Science and Technology, Government of India (Project No. SB/S1/IC-05/2014) and 'Support for Tokyotech Advanced Researchers (STAR)'. K.Y. and S.P. thank the JSPS and CSIR, India, respectively, for a Research Fellowship. We thank Tokyo Tech and IIT Madras for encouragement through MoU.

## Author contributions

K.Y., D.K.C. and M.Y. designed the work, carried out research, analysed data and wrote the paper. M.A. and S.P. were involved in the work discussion. T.K. and H.S. contributed to crystallographic analysis. M.Y. is the principal investigator. All authors discussed the results and commented on the manuscript.

## Additional information

**Competing interests:** The authors declare no competing financial interests.

