## [Peer Review File · Nature Communications]

Reviewers' comments:

Reviewer #1 (Remarks to the Author):

Review for Nature Communications manuscript NCOMMS-17-03154-T

"Polyaromatic Molecular Peanuts" by Yoshizawa and co-workers reports the synthesis of a double capsule Pd₃L₄ metallocage system that the authors suggest is a molecular version of a peanut. The system is a larger analogue of the well-known M₂L₄ anthracene based cages that have previously been developed by the Yoshizawa group. The synthesis of the new double capsule Pd₃L₄ cage is supported by NMR, MS and X-ray data and this is all completely consistent with the formation of the proposed structure. The X-ray structure has some of the usual issues associated with crystal structures of large metallocages but is of more than sufficient quality to allow the bond connectivity and overall structure of the Pd₃L₄ to be reliably determined.

The authors then examine the host-guest chemistry of the Pd₃L₄ metallocage system with a variety of organic guest molecules. With C₆₀ a cage transformation to a double cage Pd₂L₄ system is observed. The authors propose that the central Pd ion of the Pd₃L₄ capsule is expelled from the cage when two C₆₀ guest are bound within the structure. Again the provided data are consistent with this proposal. But it is remarkable, to me at least, that guest binding, through weak supramolecular forces, results in the breakage of four Pd-N "covalent" bonds. The data certainly suggest that this has happened but I wonder how this occurs?

What happens to the central Pd ion? what does it turn into [Pd(Solvent)₄]²⁺? Pd nanoparticles? This is not clear. Also the central Pd ion is the most hindered/protected so what is the mechanism for its removal. These are possibly hard questions to answer but certainly worthy of some discussion as it is intriguing that a Pd ion is lost from the cage especially when considering that the binding of other aromatic guest later in the manuscript do not cause the same Pd expulsion event. Could the authors calculate the relative energies of the Pd₂L₄:(C₆₀)₂ host guest system and the parent Pd₃L₄ metallocage to gain further insight into this interesting behaviour?

The structure of the Pd₂L₄:(C₆₀)₂ host guest system looks unstable/destabilised to me given the four lone pairs of the central pyridine ligands all point directly into the same space

Have the authors examined the conformations of the Pd₂L₄:(C₆₀)₂ host guest system (using calculations) to confirm the structure?

The authors state that the capsule 2b could be converted into Pd₂L₄:(C₆₀)₂ system but don't mention 2a does this system work in the same way and if not why not?

In a second part the manuscript the authors show that they can bind two different guests at the same time in different cavities. They make the claim that this is the first observation of this type of selective guest binding behaviour in a metallocage and while it is certainly rare I don't think it is the first time this has been observed in the literature. The Crowley paper (Ref 29) details similar heteroleptic selective guest binding in different cavities of the same multicavity system. This does not diminish the work described here but it should be mentioned more clearly in the text or the introduction.

Also Fujita and co-workers have developed many systems in which two different guests are bound with a single cavity which could be worth mentioning as well.

The heteroleptic guest binding is presumably allosteric and the binding of one guest preorganised the second cavity to fit the second guest "better". Can the authors examine this in further detail? For example what happens if they take the homoleptic host guest systems and add the second guest do they get the formation of the heteroleptic host-guest system in both cases?

The paper is well written the data well-presented and consistent with the author's conclusions. Furthermore the results are extremely interesting as such I recommend this work be strongly considered for publication in Nat. Commun. after some minor revisions examining the questions posed above.

Reviewer #2 (Remarks to the Author):

The manuscript submitted to Nat. Commun. by Chand and Yoshizawa deals with the self-assembly of metal organic cages from organic ligands and Pd(II) cations possessing two distinct molecular cavities, each large enough to encapsulate a fullerene. Binding of two fullerenes in the neighbouring cavities of the Pd₃Ligand₄ cage goes along with release of a single Pd(II) cation, most probably as a consequence of the strain that is built up in the host-guest complex.

The current work is based on a number of previous reports by the Yoshizawa group about fullerene binding single-cavity coordination cages built from anthracene-equipped ligands as well as Chands recent report on Pd₃L₄ cages featuring two distinct cavities (Chem. Eur. J. 2014, 20, 13122). The latter cages, there called 'double deckers' were already shown to encapsulate two guests in their two neighbouring cavities (albeit small anions instead of the certainly more spectacular fullerenes reported here), therefore the statement "However, there has been no report on creation of the characteristic core-shell structure at a molecular level so far" in the introduction may be questioned. The encapsulation of two C₆₀ molecules is under release of Pd is quite interesting but the asymmetric coencapsulation of two different guests is even more impressive in my opinion. What would have been nice to see were any interaction between the coencapsulated guests (C₆₀ or others) such as electron transfer. Nevertheless, I think this nice work is suitable for being published in Nat. Commun.

Comments:

- some english language polishing is needed, e.g. in "shuttling and rotating.."
- While the described system certainly resembles a peanut or likewise beans in a pod, I think that commencing the abstract with seeking a relationship with the shape of plant products will be misleading for the reader and should be removed. The next sentence is sufficient to make the point. Same is true for the rather superfluous comments on seeds and beans in the introduction and the last sentence of the conclusion..
- "Next, it was anticipated that strong π -stacking interactions between the two polyaromatic cavities and two fullerenes as a spherical polyaromatic "bean" would give rise to a molecular peanut, accompanied by the release of a single metal hinge from the double capsule": I wonder if the authors could have really foreseen the release of the central metal cation (on the basis of which conception or modeling data?) or if this wasn't a serendipitous finding.
- I am not happy with the comparison of double-capsule 2 with the fullerene dimer as shown in Fig. 3. Besides the confusion that depicting the fullerene dimer might suggest encapsulation (or its formation in the cavity) to the reader that doesn't fully immerse in the manuscript, I do not really see the point of this comparison since the covalently closed C₁₂₀ is certainly a very different kind of chemical compound with a totally different encapsulation chemistry than the reported self-assembly.
- I acknowledge the ESI TOF results indicating the loss of Pd(II) upon C₆₀ binding but one could question the NMR evidence based on the central pyridine's upfield shift. This may also stem from conformational changes in this region. Have any attempts been made to analyze (or chemically convert) the presence of the released free Pd cations?
- what is the cause of the release of Pd upon C₆₀ encapsulation? If it is strain, can this be deduced by molecular modeling in a quantitative way?
- page 8: is the homoleptic complex as written "(4b/(4b)₂)@2b" or is it "((4b)₂/(4b)₂)@2b"?
- I wonder that no mention is made on the possible positive or negative cooperativity of the encapsulation events in the two pockets?

For the comments of Reviewer #1:

The synthesis of the new double capsule Pd3L4 cage is supported by NMR, MS and X-ray data and this is all completely consistent with the formation of the proposed structure. The X-ray structure has some of the usual issues associated with crystal structures of large metallocages but is of more than sufficient quality to allow the bond connectivity and overall structure of the Pd3L4 to be reliably determined. The authors then examine the host-guest chemistry of the Pd3L4 metallocage system with a variety of organic guest molecules. With C60 a cage transformation to a double cage Pd2L4 system is observed. The authors propose that the central Pd ion of the Pd3L4 capsule is expelled from the cage when two C60 guest are bound within the structure. Again the provided data are consistent with this proposal.

We appreciate having very positive evaluation from Reviewer #1 on our molecular design, synthesis, and structural determination in this report.

1) But it is remarkable, to me at least, that guest binding, through weak supramolecular forces, results in the breakage of four Pd-N “covalent” bonds. The data certainly suggest that this has happened but I wonder how this occurs? What happens to the central Pd ion? what does it turn into [Pd(Solvent)4]2+? Pd nanoparticles? This is not clear.

We added “through the cleavage of the Pd(II)-pyridine coordination bonds” to the main text (page 4) to emphasize the cleavage of four Pd-N “non-covalent” bonds. Pd(II)-pyridine coordination bonds are usually reversible under elevated temperatures (e.g., >80 °C). In addition, the dissociated Pd(II) ion can be stabilized in coordinative solvents (DMSO) due to the solvation.

2) Also the central Pd ion is the most hindered/protected so what is the mechanism for its removal. These are possibly hard questions to answer but certainly worthy of some discussion as it is intriguing that a Pd ion is lost from the cage especially when considering that the binding of other aromatic guest later in the manuscript do not cause the same Pd expulsion event.

The central Pd(II)-pyridine bonds are protected but *strained*. As a proposed mechanism, we added “Heating is essential for the (C₆₀)₂@3b formation. Presumably the elevated temperature helps the dissociation of the central Pd(II) ion, the partial or full disassembly of the double capsule, and the dissolution of C₆₀ in DMSO” to the main text (page 5).

3) Could the authors calculate the relative energies of the Pd2L4:(C60)2 host guest system and the parent Pd3L4 metallocage to gain further insight into this interesting behaviour? The structure of the Pd2L4:(C60)2 host guest system looks unstable/destabilised to me given the four lone pairs of the central pyridine ligands all point directly into the same space. Have the authors examined the conformations of the Pd2L4:(C60)2 host guest system (using calculations) to confirm the structure?

According to this reviewer’s comment, we added the calculated energies of “(C₆₀)₂@Pd₂L₄ + Pd(pyridine)₄”, “Pd₃L₄ + 2•C₆₀ + 4•pyridine”, and “(C₆₀)₂@Pd₃L₄ + 4•pyridine” states in the gas phase to the revised SI (Supplementary Fig. 63). The preliminary theoretical result indicates that the “(C₆₀)₂@Pd₂L₄ + Pd(pyridine)₄” state is more stable than the others, which is consistent with the experimental data. Actually, the (C₆₀)₂@Pd₂L₄ structure is stable enough at elevated temperatures (up to 110 °C), under high dilution conditions (up to 5 μM), and even in the presence of excess Pd(II) ions due to multiple host-guest π-stacking interactions, as already mentioned in the main text. N•••N distance (5.3 Å) between the opposing central pyridine rings of (C₆₀)₂@Pd₂L₄ is not close enough to interact each other.

4) The authors state that the capsule 2b could be converted into Pd2L4:(C60)2 system but don't mention 2a does this system work in the same why and if not why not?

Reactivity of double capsules **2a** and **2b** is same but the solubility of the corresponding host-guest products is different. Thus we revealed the detailed host-guest structures using well-soluble **2b**. We added the ¹H NMR and ESI-TOF MS spectra of (C₆₀)₂@**3a** (prepared from double capsule **2a** and C₆₀) to the revised SI (Supplementary Figs 35 and 40B).

5) In a second part the manuscript the authors show that they can bind two different guests at the same time in different cavities. They make the claim that this is the first observation of this type of selective guest binding behaviour in a metallocage and while it is certainly rare I don't think it is the first time this has been observed in the literature. The Crowley paper (Ref 29) details similar heteroleptic selective guest binding in different cavities of the same multicavity system. This does not diminish the work described here but it should be mentioned more clearly in the text or the introduction. Also Fujita and co-workers have developed many systems in which two different guests are bound with a single cavity which could be worth mentioning as well.

According to this reviewer's comment, we added "The unusual heteroleptic encapsulation most probably arises from the cooperative changes in volume of the linked cavities upon guest encapsulation, which is mechanistically different from a previous heteroleptic binding in three cavities with different binding sites²⁹" to the main text (page 6). There are a lot of nice reports (>10 papers) on the encapsulation of two different guests in a single host cavity. Thus, we added the representative reports to the reference part (31-35) and the following sentence: "This heteroleptic encapsulation is also in sharp contrast to the pairwise encapsulation of two different guests in a single host cavity³¹⁻³⁵".

6) The heteroleptic guest binding is presumably allosteric and the binding of one guest preorganised the second cavity to fit the second guest "better". Can the authors examine this in further detail? For example what happens if they take the homoleptic host-guest systems and add the second guest do they get the formation of the heteroleptic host-guest system in both cases?

The 1:1:2 host-guest-guest' complex (**4a**/(**4b**)₂)@**2b** is a thermodynamically favorable product. Thus, the order of the guest addition to **2b** does not effect the formation of the final product.

The paper is well written the data well-presented and consistent with the author's conclusions. Furthermore the results are extremely interesting as such I recommend this work be strongly considered for publication in Nat. Commun. after some minor revisions examining the questions posed above.

We sincerely thank Reviewer #1 for his or her strong recommendation for this publication.

For the comments of Reviewer #2:

The current work is based on a number of previous reports by the Yoshizawa group about fullerene binding single-cavity coordination cages built from anthracene-equipped ligands as well as Chand's recent report on Pd₃L₄ cages featuring two distinct cavities (Chem. Eur. J. 2014, 20, 13122). The latter cages, there called 'double deckers' were already shown to encapsulate two guests in their two neighbouring cavities (albeit small anions instead of the certainly more spectacular fullerenes reported here), therefore the statement "However, there has been no report on creation of the characteristic core-shell structure at a molecular level so far" in the introduction may be questioned.

The encapsulation of two C₆₀ molecules is under release of Pd is quite interesting but the asymmetric coencapsulation of two different guests is even more impressive in my opinion. What would have been nice to see were any interaction between the coencapsulated guests (C₆₀ or others) such as electron transfer. Nevertheless, I think this nice work is suitable for being published in Nat. Commun.

We appreciate having very positive evaluation from Reviewer #2 on our report. We would like to emphasize "there has been no report on creation of the characteristic "peanut-like" core-shell structure on a nanoscale so far" in the introduction. The shape, size, and components of the peanut-shaped (C₆₀)₂@Pd₂L₄ structure is quite different from those of the previous Pd₃L₄ 'double decker' reported by Chand (the coauthor). Exploration of unique interactions between coencapsulated guests is our next project.

1) some english language polishing is needed, e.g. in "shuttling and rotating.."

The revised manuscript has been polished by a native English speaker.

2) While the described system certainly resembles a peanut or likewise beans in a pod, I think that commencing the abstract with seeking a relationship with the shape of plant products will be misleading for the reader and should be removed. The next sentence is sufficient to make the point. Same is true for the rather superfluous comments on seeds and beans in the introduction and the last sentence of the conclusion..

According to these reviewer's comments, we shorten the first sentence of the abstract but did not fully remove it for the general readers. On the other hand, all of "beans" and "pod" in the abstract and conclusion were replaced by "molecules" and "shell", respectively.

3) "Next, it was anticipated that strong pi-stacking interactions between the two polyaromatic cavities and two fullerenes as a spherical polyaromatic "bean" would give rise to a molecular peanut, accompanied by the release of a single metal hinge from the double capsule": I wonder if the authors could have really foreseen the release of the central metal cation (on the basis of which conception or modeling data?) or if this wasn't a serendipitous finding.

Preliminary molecular modeling studies had been suggested that the (C₆₀)₂@Pd₃L₄ structure provides strained central pyridine-Pd(II) bonds (See the Supplementary Fig. 63). Thus, we anticipated that the cleavage of the pyridine-Pd(II) bonds could lead to the formation of the (C₆₀)₂@Pd₂L₄ structure.

4) I am not happy with the comparison of double-capsule 2 with the fullerene dimer as shown in Fig. 3. Besides the confusion that depicting the fullerene dimer might suggest encapsulation (or its formation in the cavity) to the reader that doesn't fully immerse in the manuscript, I do not really see the point of this comparison since the covalently closed C₁₂₀ is certainly a very different kind of chemical compound with a totally different encapsulation chemistry than the reported self-assembly.

According to this reviewer's comment, we removed the structures of the fullerene dimer in Fig. 3 and the sentence of the comparison between their cavity volumes.

5) *I acknowledge the ESI TOF results indicating the loss of Pd(II) upon C60 binding but one could question the NMR evidence based on the central pyridine's upfield shift. This may also stem from conformational changes in this region. Have any attempts been made to analyze (or chemically convert) the presence of the released free Pd cations?*

The central pyridine ring on the W-shaped ligand can take only one conformation due to the steric hindrance. The dissociated Pd(II) ions seem to be stabilized in coordinative solvents (DMSO) due to the solvation. Excess free Pd(II) ions did not interact with molecular peanut $(C_{60})_2@3b$.

6) *what is the cause of the release of Pd upon C60 encapsulation? If it is strain, can this be deduced by molecular modeling in a quantitative way?*

N•••N distances between the opposing central pyridine rings of double capsule **2a'** (X-ray structure) and molecular peanut $(C_{60})_2@3b$ (optimized structure) are 4.0 and 5.3 Å, respectively. This distance change most probably causes the release of the central Pd(II) ions. We had calculated the energies of " $(C_{60})_2@Pd_2L_4 + Pd(pyridine)_4$ ", " $Pd_3L_4 + 2C_{60} + 4pyridine$ ", and " $(C_{60})_2@Pd_3L_4 + 4pyridine$ " states in the gas phase (Supplementary Fig. 63). The preliminary theoretical data had supported the present experimental data.

7) *page 8: is the homoleptic complex as written "(4b/(4b)2)@2b" or is it "(4b)2/(4b)2)@2b"?*

The ESI-TOF MS analysis revealed the selective formation of $(4b/(4b)_2)@2b$.

8) *I wonder that no mention is made on the possible positive or negative cooperativity of the encapsulation events in the two pockets?*

I modified the related sentence as follows: "*The unusual heteroleptic encapsulation most probably arises from the cooperative changes in volume of the linked cavities upon guest encapsulation*" (page 6).

REVIEWERS' COMMENTS:

Reviewer #1 (Remarks to the Author):

Manuscript#: NCOMMS-17-03154A

This is an excellent paper and the authors have carried out all the revisions requested by the reviewers

Therefore I recommend the work be published as it will be of interest to many workers in the areas of metallosupramolecular and host-guest chemistry

Reviewer #2 (Remarks to the Author):

In the revised version, the authors have addressed all raised points adequately and the manuscript has improved significantly in quality. I suggest acceptance of the revised manuscript as it is.